# Bifidobacterium longum BORI inhibits pain behavior and chondrocyte death, and attenuates osteoarthritis progression

Dong Keon Oh[1,☯,¤a], Hyun Sik Na[1,☯,¤a], Joo Yeon Jhun[1,¤a], Jeong Su Lee[1,¤a], In Gyu Um[1,¤a], Seung Yoon Lee[1,¤a], Myeong Soo Park[2], Mi-La Cho[1,¤a,¤b,‡,*], Sung-Hwan Park[3,‡,*]

**1** Rheumatism Research Center, Catholic Research Institute of Medical Science, The Catholic University of Korea, Seoul, Korea, **2** Research Center, BIFIDO Co., Ltd., Hongcheon, South Korea, **3** Division of Rheumatology, Department of Internal Medicine, Seoul St. Mary's Hospital, College of Medicine, Catholic University of Korea, Seoul, Korea

☯ These authors contributed equally to this work.
¤a Current address: Department of Biomedicine & Health Sciences, College of Medicine, The Catholic University of Korea, Seoul, Korea
¤b Current address: Department of Medical Lifescience, College of Medicine, The Catholic University of Korea, Seoul, Korea
‡ MLC and SHP also contributed equally to this work.
* iammila@catholic.ac.kr (MLC); rapark@catholic.ac.kr iammila@catholic.ac.kr (SHP)

**Data Availability Statement:** Data available on figshare at DOI: 10.6084/m9.figshare.23354147

**Funding:** This research was supported by a grant of the Food Industry Promotional Agency of Korea

## Abstract

Osteoarthritis (OA), the most common form of arthritis, is characterized by pain and cartilage damage; it usually exhibits gradual development. However, the pathogenesis of OA remains unclear. This study was undertaken to improve the understanding and treatment of OA. OA was induced in 7-week-old Wistar rats by intra-articular injection of monosodium iodoacetate (MIA); subsequently, the rats underwent oral administration of *Bifidobacterium longum* BORI (*B.* BORI). The effects of *B.* BORI were examined in chondrocytes and an MIA-induced OA rat model. In the rats, *B.* BORI-mediated effects on pain severity, cartilage destruction, and inflammation were recorded. Additional effects on mRNA and cytokine secretion were analyzed by quantitative polymerase chain reaction and enzyme-linked immunosorbent assay. Paw withdrawal threshold, paw withdrawal latency, and weight-bearing assessments revealed that pain severity in MIA-induced OA rats was decreased after *B.* BORI treatment. Histopathology analyses and three-dimensional surface renderings of rat femurs from micro-computed tomography images revealed cartilage protection and cartilage loss inhibition effects in *B.* BORI-treated OA rats. Immunohistochemical analyses of inflammatory cytokines and catabolic markers (e.g., matrix metalloproteinases) showed that the expression levels of both were reduced in tissue from *B.* BORI-treated OA rats. Furthermore, *B.* BORI treatment decreased the expression levels of the inflammatory cytokine monocyte chemoattractant protein-1 and inflammatory gene factors (e.g., inflammatory cell death markers) in chondrocytes. The findings indicate that oral administration of *B.* BORI has therapeutic potential in terms of reducing pain, progression, and inflammation in OA.

and this research was supported by a grant of the Korea Health Technology R&D Project through the Korea Health Industry Development Institute (KHIDI), funded by the Ministry of Health & Welfare, Republic of Korea (grant number HI20C1496). The funders had no role in study design, data collection and analysis, decision to publish, or preparation of the manuscript.

**Competing interests:** The authors have declared that no competing interests exist.

## Introduction

Osteoarthritis (OA), the most common form of arthritis, mainly occurs in older adults [1]. It usually involves the onset of pain and stiffness in the joints upon awakening or after periods of inactivity; pathological symptoms include degradation of cartilage and bone, formation of bone (i.e., osteophytes), and inflammation of the synovial membrane, all of which become increasingly severe over time [2]. OA can be caused by cartilage degradation, but studies in the past decade have shown that OA involves the release of numerous proinflammatory cytokines and matrix metalloproteinases (MMPs) from cartilage, bone, and synovium. These inflammatory agents cause cartilage destruction, highlighting the complex underlying mechanisms of OA [3].

The regulation of chronic inflammation through gut microbiota-focused modulation has been proposed as a new treatment option for OA [4, 5]. Probiotics stabilize intestinal flora, suppress harmful bacteria, activate immunity, exhibit anticancer activity, and alleviate lactose intolerance. The immunomodulatory effects depend on the method of administration, class of immune disorder, severity of disease, and type of probiotic [6]. Representative probiotics include *Lactobacillus* and *Bifidobacterium*. *Lactobacillus* is classified as a generally regarded as safe (GRAS) microorganism and has considerable value as a probiotic [7, 8].

*Bifidobacterium* is a common bacterium in the human intestine and a dominant component of the intestinal flora. It constitutes ˜25% of the fecal microbial community in human adults and ˜80% of the community in human infants [9]. *Bifidobacterium* is anaerobic; unlike other lactic acid bacteria, it specifically produces L (+)-lactic acid, which is easily absorbed in the human body and does not cause acidosis in infants.

Additionally, *Bifidobacterium* exhibits mutual interactions with various intestinal bacterial flora and has effects on human nutrition, aging, carcinogenesis, immune function, intestinal infection, and drug metabolism [10]. *Bifidobacterium longum*, one of 32 species in the *Bifidobacterium* genus, is more effective than other members of its genus in terms of inhibiting inflammation [11, 12]. Therefore, *B. longum* may have immunomodulatory and chondroprotective effects, suggesting that it could be useful as a probiotic in the treatment of OA [13].

Here, we investigated this potential therapeutic effect by administering heat-killed *B. longum* BORI (*B*. BORI) to rats with monosodium iodoacetate (MIA)-induced OA. Monosodium iodoacetate (MIA)-induced OA rats are widely used as a human osteoarthritis model, by destroying cartilage cells, causing degeneration and inflammation of the cartilage [14]. This causes symptoms similar to those of thoracic osteoarthritis. The administration of *B*. BORI is safe and has demonstrated positive effects in terms of inhibiting inflammation in cranial nerve cells, as well as hepatic and intestinal conditions [10, 15]. Furthermore, B. BORI demonstrated the maintenance of gut microbiota balance and stability by showing effectiveness in preventing and treating rotavirus infection in infants [16]. However, no studies have confirmed the therapeutic effect of *B*. BORI on OA. We hypothesized that *B*. BORI could effectively treat arthritis, and we expect that our findings help to support future clinical studies.

## Methods

### Ethics statement

The Animal Care Committee of The Catholic University of Korea approved the experimental protocol (permit number: CUMC-2021-063-01). All animal handling procedures and protocols followed the guidelines of the Laboratory Animals Welfare Act, the Guide for the Care and Use of Laboratory Animals, and the Guidelines and Policies for Rodent Experiments provided by the Institutional Animal Care and Use Committee (IACUC) of the School of Medicine, The Catholic University of Korea.

## Preparation of bacteria

B. longum BORI was isolated from human feces of healthy infant who lived in Chuncheon, South Korea between 1995 and 1998, identified with 16S rRNA sequencing and cultured with MRS (De Man, Rogosa and Sharpe, Becton Dickson, Franklin Lakes, NJ, USA) broth medium at 37 ˚C. For the initial screening, lyophilized powder of each strain was used. Freeze-dried powder of B. longum BORI was provided by BIFIDO Co. Ltd (Hongchun, Korea).

## Induction of OA in rats and treatment with *B*. BORI

MIA-treated rats were randomized into experimental groups. OA was induced in 7-week-old male Wistar rats (n = 6). The experiment was performed two times. After rats had been anesthetized with isoflurane, they underwent injection of MIA (3 mg in 50 μL; Sigma-Aldrich, St. Louis, MO, USA) through a 26.5-G needle that was inserted via the patellar ligament into the intra-articular space of the right knee. *B*. BORI and celecoxib were administered from the third day after MIA injection; *B*. BORI was administered orally after it had been dissolved in saline at a concentration of $1 \times 10^9$ colony-forming units per rat, whereas celecoxib was administered orally at a concentration of 30 mg/kg in 0.5% carboxymethyl cellulose solution. All medications were administered three times per week, and all rats were sacrificed on the 21st day after injection of MIA.

## Assessment of pain behavior

MIA-treated rats were randomly assigned to experimental groups. Nociception measurements were made using a dynamic plantar aesthesiometer (Ugo Basile, Gemonio, Italy), an automated version of the von Frey hair assessment procedure. This experiment was conducted on a metal mesh surface in an acrylic chamber, within a temperature-controlled environment (20–26˚C). Each rat was allowed to acclimate for 15 min before measurements. Then, the aesthesiometer was placed under the rat using an adjustable angled mirror; a stimulating microfilament (0.5 mm diameter) was attached for stimulation of the plantar surface of the hind paw. When the instrument was activated, a fine plastic monofilament advanced at a constant speed and touched the paw in the proximal metatarsal region. The filament exerted a gradually increasing force on the plantar surface, beginning below the detection threshold and increasing until the stimulus became painful, as indicated by the rat's withdrawal of its paw. The force (in g) required to elicit a paw withdrawal reflex was recorded automatically. The maximum force of 50 g and a ramp speed of 25 s were used for all aesthesiometer tests.

## Weight-bearing measurement

An incapacitance meter (IITC Life Science, CA, USA) was used for weight-bearing measurements in MIA-treated rats. After a rat had acclimated on the acrylic holder for 5 min, measurements were recorded when both feet were fixed on each weight. This experiment was repeated three times in the same manner; the weights of the unguided and guided legs were determined, then used to find the % value via comparison between legs with and without OA.

## Immunohistochemical analyses

Paraffin-embedded sections were deparaffinized and rehydrated in graded concentrations of ethanol. The sections were then incubated at 4˚C with antibodies against interleukin (IL)-1β (1:400 dilution, NB600-633, Novus Biologicals), leukotriene B4 receptor (LTB4r; 1:100 dilution, BS-2654R, Thermo Fisher Scientific), prostaglandin E2 (PGE2; 1:250 dilution, bs-2639R, Bioss Antibodies), matrix metalloproteinase-9 (MMP9; 1:100 dilution, MA5-32705, Thermo

Fisher Scientific), and matrix metalloproteinase-13 (MMP13; 1:300 dilution, ab9012, Abcam). Next, the sections were incubated with the respective secondary biotinylated antibodies, then incubated for 30 min with a streptavidin–peroxidase complex. Reaction products were developed using 3,3-diaminobenzidine chromogen (Dako, Carpinteria, CA, USA). The all IHC images were obtained each rat, and showing representative images. The percentage of positive cells showing in each image was measured by Image J.

## Quantitative polymerase chain reaction

Total RNA was extracted using TRI Reagent (Molecular Research Center, Cincinnati, OH, USA), then used for complementary DNA (cDNA) synthesis with a high-capacity cDNA reverse transcription kit (Applied Biosystems, Foster City, CA, USA) in accordance with the manufacturer's instructions. Polymerase chain reaction amplification was performed using LightCycler FastStart DNA Master SYBR Green I (TaKaRa, Shiga, Japan) and a LightCycler 2.0 instrument (software version 4.0; Roche Diagnostics, Indianapolis, IN, USA). The following primer pairs were used: β-actin forward, 5'-CAT GTA CGT TGC TAT CCA GGC-3'; β-actin reverse, 5'-CTC CTT AAT GTC ACG CAC GAT-3'; caspase-1 forward, 5'-TTT CCG CAA GGT TCG ATT TTC A-3'; caspase-1 reverse, 5'-GGC ATC TGC GCT CTA CCA TC-3'; receptor-interacting protein kinase (RIPK)3 forward, 5'-ATG TCG TGC GTC AAG TTA TGG-3'; and RIPK3 reverse, 5'-CGT AGC CCC ACT TCC TAT GTT G-3'.

## Enzyme-linked immunosorbent assay

The concentrations of monocyte chemoattractant protein-1 (MCP-1) in culture supernatants were measured by the DuoSet enzyme-linked immunosorbent assay kit (R&D Systems, Minneapolis, MN, USA) in 96-well plates (Nunc, Roskilde, Denmark) that had been coated with anti-human MCP-1 capture antibodies by overnight incubation at 4°C. After incubation, the plates were blocked for 2 h at room temperature with phosphate-buffered saline containing 1% bovine serum albumin and 0.05% Tween 20. Cell culture supernatants were added to the plates and incubated at room temperature for 2 h. Subsequently, the plates were washed, detection antibodies were added, and the reaction mixtures were incubated for 2 h at room temperature. Then, the plates were washed and incubated with streptavidin–horseradish peroxidase for 20 min. After an additional wash step, plates were incubated with substrate solution for 20 min before the addition of stop solution. The results were analyzed by measurement of absorption at 405 nm.

## Evaluation via micro-computed tomography (CT) scanning

Micro-CT imaging and analyses of femur volume and surface were performed using a benchtop cone-beam type *in vivo* animal scanner (μCT 35; SCANCO Medical, Bruttisellen, Switzerland). Rats were imaged at 70 kVp and 141 μA using a 0.5-mm-thick aluminum filter, pixel size of 8.0 μm, and rotation step of 0.4°. Cross-sectional images were reconstructed to 2000 × 1335 pixels in each scan using a filtered back-projection algorithm (Nrecon software, Bruker micro CT, Kontich, Belgium).

## OA chondrocyte analyses

The Ethics Committee of Seoul St. Mary's Hospital (Seoul, Korea) provided approval for all chondrocyte experiments (permit numbers: KC21SISI0337). Human OA chondrocytes were incubated in Dulbecco's modified Eagle medium containing 10% fetal bovine serum. After

three passages, the chondrocytes were cultured with *B*. BORI (10 μg/mL) and IL-1β (20 ng/mL) for 24 h and 48 h.

## Statistical analysis

Statistical analysis was performed using GraphPad Software (version 5.01, San Diego, CA, USA). Comparisons among ≥ 3 groups were conducted by one-way analysis of variance; differences were confirmed via Bonferroni post hoc tests. Comparisons between two groups were conducted by two-tailed nonparametric Mann–Whitney U tests. All data are shown as means ± standard errors of the mean, and p-values < 0.05 were considered statistically significant.

## Results

### *B*. BORI suppresses pain in MIA-induced OA rats

To investigate the effects of oral administration of *B*. BORI in MIA-induced OA rats, we examined the paw withdrawal threshold (PWT), paw withdrawal latency (PWL), and weight-bearing. PWT and PWL were higher in the *B*. BORI-treated group than in the vehicle group (Fig 1A and 1B). Additionally, weight-bearing was significantly greater in the *B*. BORI-treated group than in the vehicle group (Fig 1C). These results suggest that oral administration of *B*. BORI reduced pain in OA rats.

### *B*. BORI protects against cartilage destruction in MIA-induced OA rats

We performed histological analysis using safranin O staining to determine whether orally administered *B*. BORI exhibited chondroprotective effects in MIA-induced OA rats. The Osteoarthritis Research Society International and total Mankin scores were low in the *B*. BORI-treated group (Fig 2A). Micro-CT imaging-based three-dimensional reconstruction and analysis of the subchondral bone region in MIA-induced OA rats showed that the femurs of *B*. BORI-treated rats were less degraded and more compact. Quantitative micro-CT images revealed that the bone surface (%) and object volume/total volume of the femur were higher in the *B*. BORI-treated group (Fig 2B). These results indicated that oral administration of *B*. BORI inhibited OA progression in our rat model.

### *B*. BORI modulates inflammatory mediator levels in the synovium of MIA-induced OA rats

Immunohistochemical staining was used to assess the expression of inflammatory mediators in the joint synovium of MIA-induced OA rats. The expression of IL-1β, an inflammatory cytokine, was lower in the *B*. BORI-treated group than in the vehicle group. Additionally, the level of LTB4r, which recruits immune cells, and PGE2, which regulates inflammatory cytokine activity in various immune cells, were decreased in the *B*. BORI-treated group (Fig 3). These results suggest that the administration of *B*. BORI protects against joint destruction by inhibiting the activation and recruitment of inflammatory cytokines and immune cells.

### *B*. BORI modulates catabolic factor levels in the synovium of MIA-induced OA rats

Immunohistochemical staining was performed to confirm whether *B*. BORI exhibits chondroprotective effects in an MIA-induced OA environment. The expression levels of MMP9 and MMP13, which are involved in cartilage destruction, were lower in the *B*. BORI-treated group

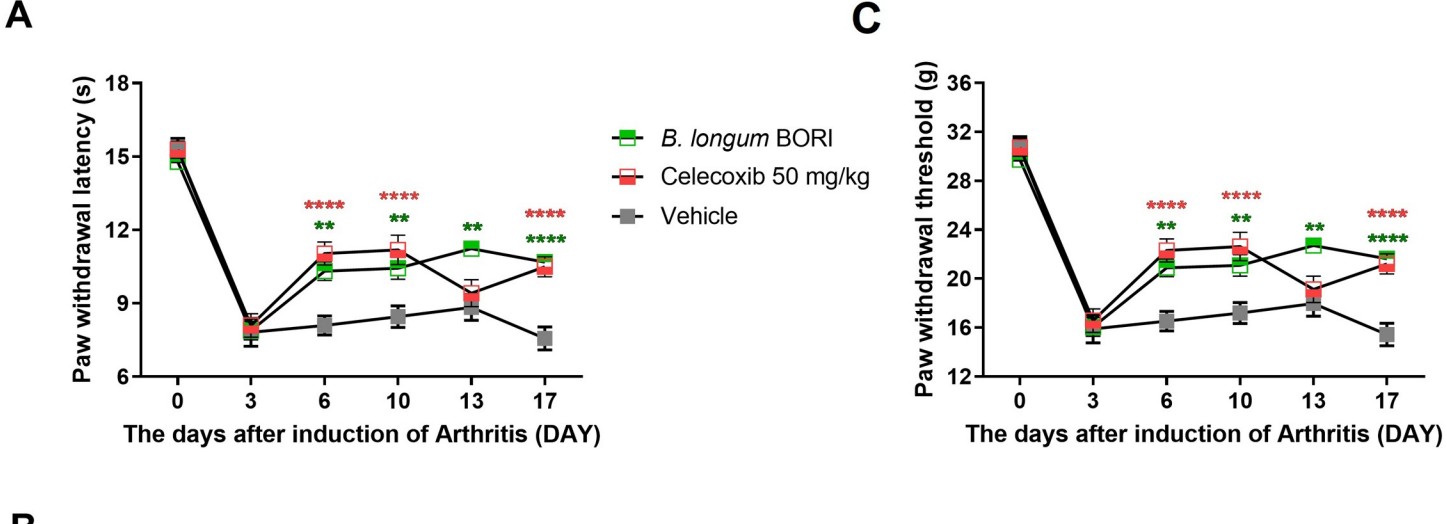

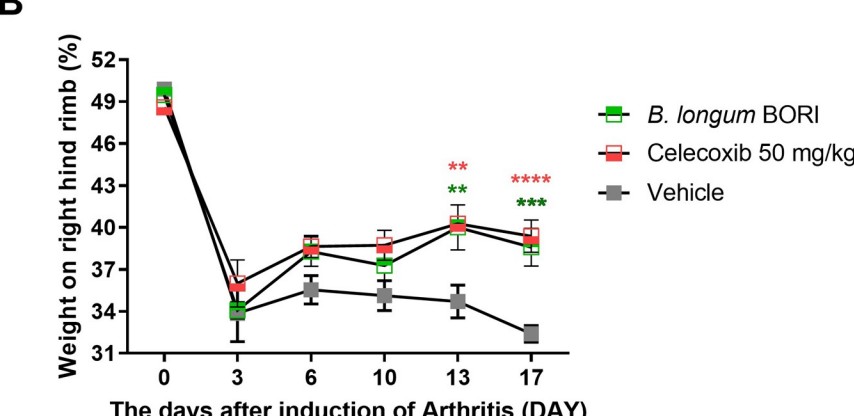

**Fig 1. *B. longum* BORI (*B*. BORI) showed a therapeutic effect in OA rats.** (**A, C**) Pain thresholds were determined by analyses of PWL (time) and PWT (threshold) in MIA-induced OA rats treated with vehicle and *B*. BORI, respectively, until day 21. (**B**) Weight-bearing was examined in all groups (n = 6 per group) until day 21. Data are shown as means ± standard errors of the mean (** $p < 0.01$ and **** $p < 0.0001$).

than in the vehicle group (Fig 4A), suggesting that *B*. BORI administration protects against cartilage destruction and the loss of type II collagen.

## *B*. BORI regulates inflammatory activity and inflammatory cell death in chondrocytes

To characterize the anti-inflammatory effects of *B*. BORI, we assessed the secretion of factors related to inflammatory activity and inflammatory cell death after the induction of an inflammatory environment using IL-1β. The level of MCP-1 secretion was decreased (Fig 5A), suggesting that *B*. BORI reduces inflammatory activity in an inflammatory environment. We also compared the expression levels of factors associated with inflammatory cell death. The levels of RIPK3, a marker of necroptosis, and caspase-1, a marker of pyroptosis, were reduced upon treatment with *B*. BORI (Fig 5B). These decreases in expression indicate that the chondroprotective effects of *B*. BORI are mediated by the inhibition of inflammatory cell death.

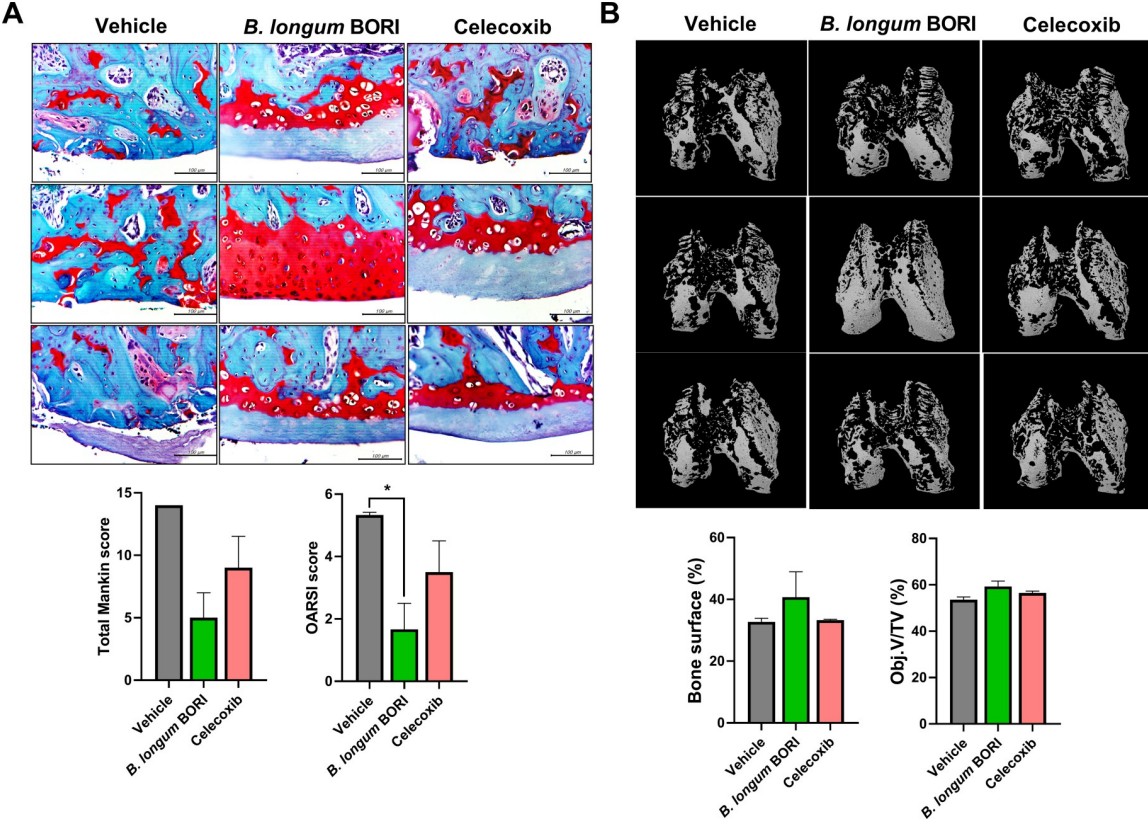

**Fig 2. *B. longum* BORI (*B*. BORI) reduces bone and cartilage erosion in MIA-induced OA rats.** Rats were injected with 3 mg of MIA. After OA induction, rats were administered *B*. BORI at $1 \times 10^9$ colony-forming units per rat. Rats were sacrificed 21 days after OA induction and joint tissues were collected. (**A**) Samples from three rats per group were stained with safranin O to determine the Osteoarthritis Research Society International (OARSI) and Mankin scores. (**B**) Femur samples were scanned via micro-computed tomography (μCT 35; SCANCO Medical, Zurich, Switzerland). Object volume (Obj.V)/total volume (TV) and bone surface were analyzed using NRecon software. Data are shown as means ± standard errors of the mean (* $p < 0.05$).

## Discussion

*B*. BORI is known for its anti-inflammatory effects, but its therapeutic effects in the context of OA have not been explored [10, 15]. In the present study, *B*. BORI reduced cartilage destruction in a rat model of MIA-induced OA, and its anti-inflammatory effects protected against OA progression. The reduced expression levels of inflammatory factors, along with the inductions of anabolic and chondrogenic transcription factors in chondrocytes, highlight its potential for use in the treatment of OA patients.

Investigations of probiotics and prebiotics to treat various diseases are underway. Probiotic treatment alters the intestinal environment by modifying the gut microbiota; such alterations can ameliorate infectious and autoimmune diseases, allergies, and Alzheimer's disease [17]. OA is a form of chronic low-grade inflammation, and the therapeutic effects of probiotics may be mediated through a gut–joint axis. Some studies have shown differences in gut flora between rheumatoid arthritis and OA, highlighting the importance of gut flora in autoimmune diseases [18].

Although the mechanisms that underlie *Enterobacteriaceae*-related autoimmune diseases are not fully understood, various metabolites (e.g., short-chain fatty acids) are presumed to exhibit anti-inflammatory effects [19–21]. Among the short-chain fatty acids, acetate, propionate, and butyrate are important metabolites involved in the maintenance of intestinal

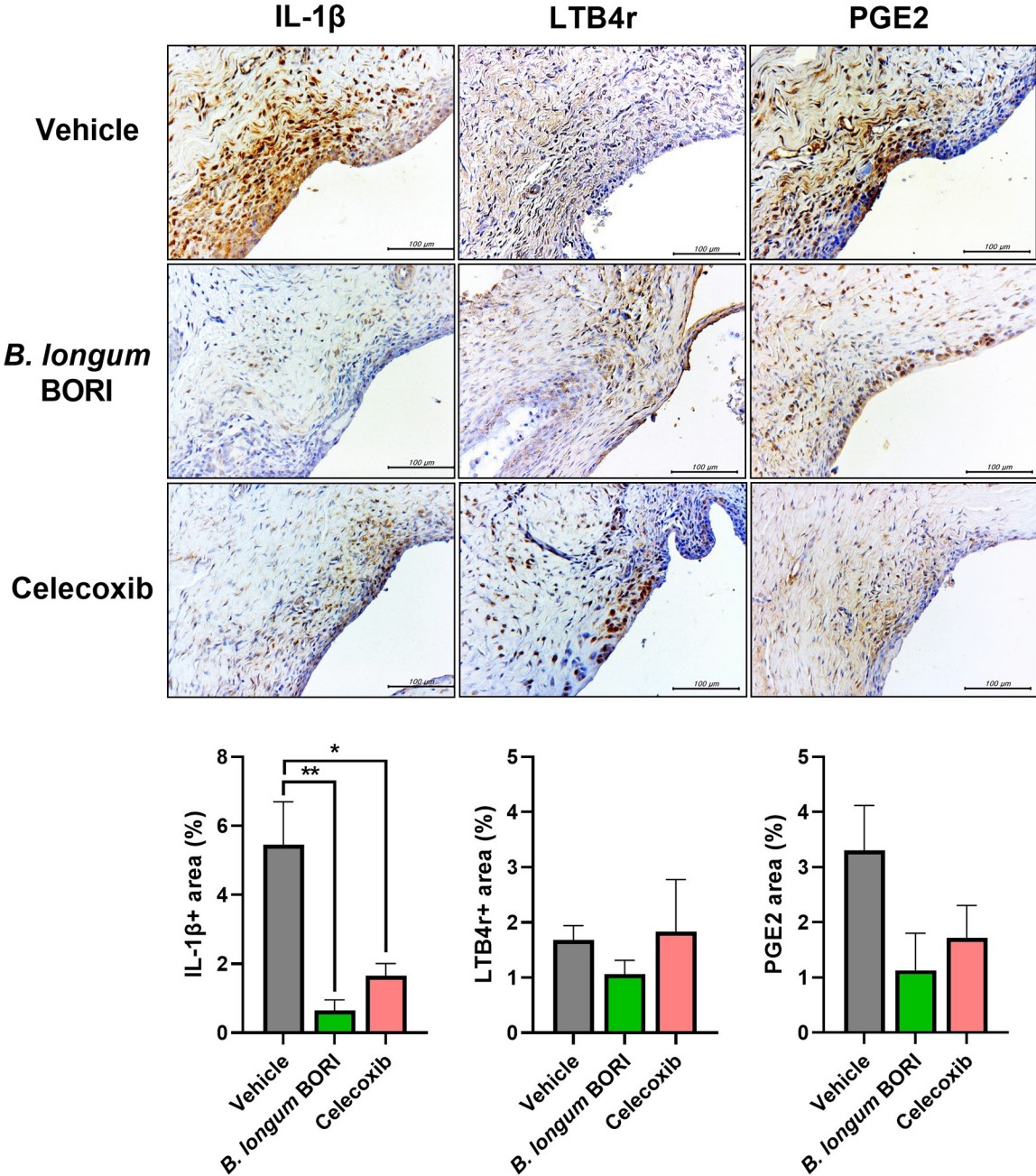

**Fig 3. Representative images of immunohistochemical staining for inflammatory mediators in the joint synovium of MIA-induced OA rats treated with B. BORI.** Bar graphs show mean numbers of IL-1β, LTB4r, and PGE2-positive cells in the joint synovium. Data are shown as means ± standard errors of the mean (* p < 0.05 and ** p < 0.01).

homeostasis; they have beneficial effects in autoimmune diseases, such as OA [22]. Acetate increases polyclonal antibody responses *in vivo* [23], and butyrate prevents cartilage destruction by reducing the degradation of type 2 collagen [24, 25]. Oral administration of the bacterium *Lactobacillus rhamnosus* led to the production of acetate, which was converted to acetyl-CoA and then used to produce butyrate [26, 27]. Oral administration *of Lactobacillus casei* Shirota reduced the levels of high-sensitivity C-reactive protein in OA patients [28], and oral

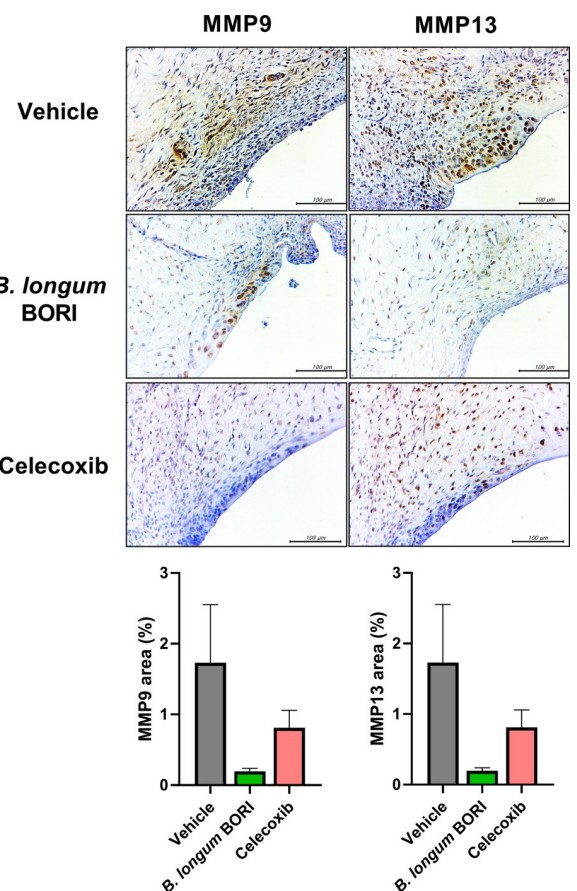

**Fig 4. Representative images of immunohistochemical staining for anabolic/catabolic factors in the joint synovium of MIA-induced OA rats treated with *B*. BORI.** Bar graphs show mean numbers of MMP9 and MMP13-positive cells in the joint synovium. Data are shown as means ± standard errors of the mean (* $p < 0.05$, ** $p < 0.01$, and *** $p < 0.001$).

administration of *Streptococcus thermophiles* inhibited disease progression in OA patients [29]. Therefore, the administration of enteric bacteria can control the intestinal environment and have therapeutic effects in patients with autoimmune diseases.

*B. longum* and other *Bifidobacterium* species constitute up to 90% of intestinal bacteria in infants. *B. longum* inhibits inflammation; in the context of Alzheimer's disease, the oral administration of *B. longum* reduces the levels of inflammatory cytokine expression in brain neurons [10, 15]. In a guinea pig OA model, the oral administration *of B. longum* CBi0703 reduced structural cartilage lesions and decreased the degradation of type II collagen [30]. However, the mechanisms underlying the role of *Bifidobacterium* spp. in the treatment of OA remain unclear. The results of one study suggested that microinflammation activates Toll-like receptors that circulate lipopolysaccharide; however, the results of another study suggested that *B. longum* did not interact with Toll-like receptors [31, 32]. Furthermore, genomic sequencing of *Bifidobacterium* spp. identified a potential eukaryotic serine protease inhibitor (serpin) that may be involved in their immunomodulatory activity [12]; this finding supports the potential for treatment using *Bifidobacterium* spp. in OA.

In this study, we measured the anti-inflammatory effect of *B*. BORI (a subtype of *B. longum*) through animal experiments to determine whether it could have therapeutic effects on OA based on its ability to inhibit inflammation in cranial nerve cells [10, 15]. Because MIA-

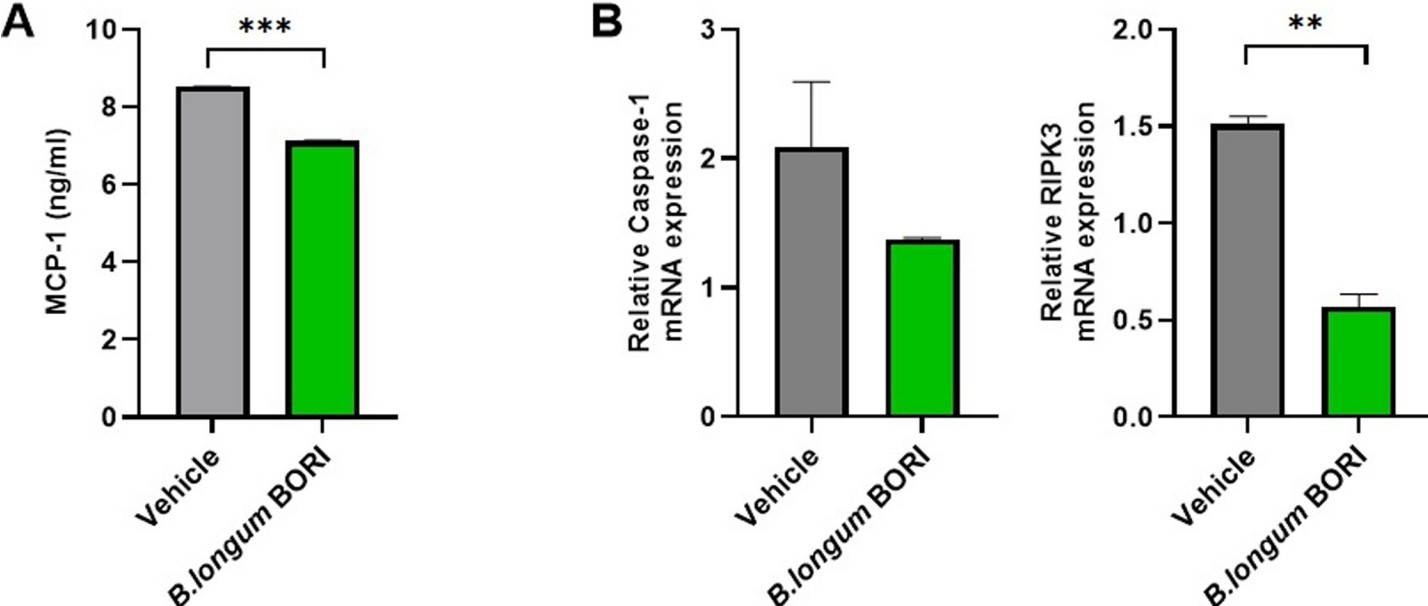

**Fig 5. *B*. BORI regulates inflammatory activity and inflammatory apoptosis in chondrocytes.** Human OA chondrocytes were cultured with *B*. BORI (10 μg/ml) and IL-1β (20 ng/ml) for 24 h. Then, supernatants and cells were collected for analysis. **(A)** MCP-1 secretion levels were measured in the supernatant of *B*. BORI-treated OA chondrocytes by enzyme-linked immunosorbent assay. **(B)** Expression levels of inflammatory cell death mediators after *B*. BORI treatment were analyzed by quantitative polymerase chain reaction. Data are shown as means ± standard errors of the mean (** $p < 0.01$ and *** $p < 0.001$).

induced OA exhibits similarities with diabetes mellitus, it is ideal for the analysis of OA in animal experiments. Here, we used MIA to induce OA in rats, then administered heat-killed *B*. BORI. The findings supported our hypothesis: studies of the differences between heat-killed and live *Bifidobacterium* spp. showed that both had the potential to modulate immune functions. The oral administration of *B*. BORI led to increased PWT, PWL, and weight-bearing capacity in MIA-induced OA rats. These indicators serve as experimental measures of load differences between legs in relation to OA-induced leg pain. Overall, we found that *B*. BORI treatment could suppress knee pain, which is regarded as an indicator of OA severity.

In OA, the levels of collagen (particularly type 2 collagen) decrease and cartilage is damaged. Therefore, we examined the degradation of type 2 collagen using safranin O staining [33, 34]. Additionally, micro-CT scans were conducted to determine the extent of cartilage damage. We confirmed reduced degradation of collagen and the suppression of cartilage damage in rats that had undergone oral administration of *B*. BORI.

Catabolic activity causes bone and cartilage destruction in OA pathogenesis, both of which are promoted by MMPs. Homeostasis involving tissue inhibitors of metalloproteinases (TIMPs), which inhibit MMPs, is essential for the regulation of cartilage tissue in OA patients [35, 36]. Immunohistochemical analysis of the joint synovium showed decreased levels of MMP9 and MMP13 in OA rats that had been treated with *B*. BORI. There were decreases in the levels of other inflammatory markers, including IL-1β (an inflammatory cytokine), LTB4r (which recruits immune cells), and PGE2 (which regulates the activity of inflammatory cytokines in immune cells), suggesting that cartilage destruction was inhibited through anti-inflammatory effects and the inhibition of catabolic responses.

Inflammatory cell death (e.g., pyroptosis, necroptosis, and ferroptosis) is induced by an inflammatory response, and chondrocytes are vulnerable to apoptosis in OA [37]. This inflammatory cell death exacerbates OA because of inflammatory cytokine release, in contrast to the

mechanism involved in apoptosis. Pyroptosis is gasdermin-mediated programmed necrotic cell death; the gasdermin family participates in pore-forming activity at the cell membrane, releasing intracellular substances (e.g., IL-1β) that stimulate the immune response [38, 39]. Because nuclear factor (NF)-κB mainly causes pyroptosis, there have been efforts to characterize the therapeutic effect of blocking this pathway in OA. For example, morroniside significantly inhibits the NF-κB signaling pathway, thereby decreasing the expression levels of NOD-, LRR- and pyrin domain-containing protein (NLRP)3 and caspase-1, leading to delayed OA progression [40]. In addition to the NF-κB-mediated inhibition of inflammatory factor expression, pyroptosis is caused by activation of the NLRP1 and NLRP3 inflammasomes. There is evidence to suggest that NLRP3 stimulation is caused by increased levels of adenosine triphosphate and lipopolysaccharide in the cartilage space, both of which contribute to inflammation [41]. Necroptosis is programmed necrotic cell death caused by RIPK1/RIPK3 and mixed lineage kinase domain-like (MLKL) under various pathological conditions [42, 43]. High levels of RIPK1/RIPK3 deactivate caspase-8, allowing the initiation of apoptosis. MLKL self-phosphorylates, then oligomerizes and translocates to the plasma membrane, generating cation channels that cause lysis of the plasma membrane [44]. Oxidative stress and mechanical stress through RIPK induce necroptosis in OA [45, 46]. Therefore, like these studies, interest of inflammatory cell death and OA are increasing [45, 47, 48]. We investigated the expression of RIPK3, a marker of necroptosis, and caspase-1, a marker of pyroptosis in chondrocytes, to determine whether *B*. BORI regulates inflammatory cell death. However, further research is needed to confirm the effects of *B*. BORI.

In conclusion, we demonstrated that *B*. BORI mediates therapeutic effects in OA via chondroprotection and pain relief. Therefore, *B*. BORI should be regarded as a potential treatment for OA

## Author Contributions

**Conceptualization:** Mi-La Cho, Sung-Hwan Park.

**Data curation:** Dong Keon Oh, Hyun Sik Na, Joo Yeon Jhun, Jeong Su Lee, In Gyu Um, Seung Yoon Lee, Mi-La Cho, Sung-Hwan Park.

**Formal analysis:** Dong Keon Oh, Hyun Sik Na, Joo Yeon Jhun, Jeong Su Lee, In Gyu Um, Seung Yoon Lee.

**Funding acquisition:** Myeong Soo Park, Mi-La Cho, Sung-Hwan Park.

**Investigation:** Dong Keon Oh, Hyun Sik Na, Joo Yeon Jhun, Jeong Su Lee, In Gyu Um, Seung Yoon Lee.

**Methodology:** Dong Keon Oh, Hyun Sik Na.

**Project administration:** Mi-La Cho, Sung-Hwan Park.

**Resources:** Mi-La Cho, Sung-Hwan Park.

**Supervision:** Dong Keon Oh, Hyun Sik Na.

**Validation:** Dong Keon Oh, Mi-La Cho.

**Visualization:** Dong Keon Oh.

**Writing – original draft:** Dong Keon Oh.

**Writing – review & editing:** Dong Keon Oh, Joo Yeon Jhun, Mi-La Cho.

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
