## [Decision Letter · Decision Letter 0]

3 Apr 2023

PONE-D-23-01788Bifidobacterium longum BORI reduces pain behavior and attenuates the progression of monosodium iodoacetate-induced osteoarthritisPLOS ONE

Dear Dr. Cho,

Thank you for submitting your manuscript to PLOS ONE. After careful consideration, we feel that it has merit but does not fully meet PLOS ONE’s publication criteria as it currently stands. Therefore, we invite you to submit a revised version of the manuscript that addresses the points raised during the review process.

ACADEMIC EDITOR:This is a well written manuscript but some major improvements are required as indicated by the reviewer 1. The authors are suggested to correct the typo errors in the revised version.

We look forward to receiving your revised manuscript.

Kind regards,

Win Min Oo

Academic Editor

PLOS ONE

“This research was supported by a grant of the Food Industry Promotional Agency of Korea and this research was supported by a grant of the Korea Health Technology R&D Project through the Korea Health Industry Development Institute (KHIDI), funded by the Ministry of Health & Welfare, Republic of Korea (grant number HI20C1496)”

“No”

Reviewers' comments:

Reviewer's Responses to Questions

**Comments to the Author**

1. Is the manuscript technically sound, and do the data support the conclusions?

Reviewer #1: Yes

2. Has the statistical analysis been performed appropriately and rigorously? 

Reviewer #1: Yes

3. Have the authors made all data underlying the findings in their manuscript fully available?

Reviewer #1: Yes

4. Is the manuscript presented in an intelligible fashion and written in standard English?

Reviewer #1: Yes

5. Review Comments to the Author

Reviewer #1: Bifidobacterium longum BORI reduces pain behavior and attenuates the progression of monosodium iodoacetate-induced osteoarthritis

Reviewer comments

This paper submitted by the authors investigated whether B. longum BORI modulates arthritic symptoms and accompanying pain symptoms in an osteoarthritis model. Probiotics such as B. longum are currently being investigated and published for their disease-modulating effects in various inflammatory diseases. The results presented in this paper will add to our knowledge of probiotic therapies that can inhibit articular cartilage damage and pain in osteoarthritis models. I have several questions regarding this study.

1) In Background section authors present the topic well and with appropriate reference however little is said about the rational of their study. My suggestion is to expand on the relevance of in vitro models used, the bacteria strains they used, why they used that strains and the relevance these strains might have.

2) The author seems to have partially mentioned the number of experiments in the experiment method and results. It is desirable that the total number of experiments performed in this part be accurately reflected.

3) Why are some experiments used after killing B. longum BORI? An explanation is needed as to why dead bacteria were used.

4) In methods, "Preparation of bacteria" it is not mentioned what sample was used for bacteria isolation

5)In the same section, "Immunohistochemical staining of joint synovium showed that administration of B. longum BORI reduced the levels of IL-1b, LTB4r, PGE2, MMP9 and MMP13 (Figures 3 and 4)." . Histology in the method of Figures 3, 4 and 4 is not clear how to observe the reduction of the listed cytokines. It is necessary to add a description of how the cytokine reduction is observed.

6) The title refers only to experiments with rats. The experiment also includes the contents of the experiment in the patient's chondrocyte culture, so it would be good to mention these parts in the title.

7) In figure, B. longum BORI was not mentioned in Italics. It should be corrected throughout the Figure.

8) The authors should better check the manuscripts for any typographical errors.

For example, LTB4 must be modified to LTB4r in figure 3.

6. PLOS authors have the option to publish the peer review history of their article (what does this mean?). If published, this will include your full peer review and any attached files.

Reviewer #1: No

---

## [Author Response · Author response to Decision Letter 0]

6 Apr 2023

Reviewer comments

This paper submitted by the authors investigated whether B. longum BORI modulates arthritic symptoms and accompanying pain symptoms in an osteoarthritis model. Probiotics such as B. longum are currently being investigated and published for their disease-modulating effects in various inflammatory diseases. The results presented in this paper will add to our knowledge of probiotic therapies that can inhibit articular cartilage damage and pain in osteoarthritis models. I have several questions regarding this study.

1) In Background section authors present the topic well and with appropriate reference however little is said about the rational of their study. My suggestion is to expand on the relevance of in vitro models used, the bacteria strains they used, why they used that strains and the relevance these strains might have.

Response: We agree with referee’s comment. We revised the introduction section (marker by red color)

Monosodium iodoacetate (MIA)-induced OA rats are widely used as a human osteoarthritis model, by destroying cartilage cells, causing degeneration and inflammation of the cartilage[14]. This causes symptoms similar to those of thoracic osteoarthritis. The administration of B. BORI is safe and has demonstrated positive effects in terms of inhibiting inflammation in cranial nerve cells, as well as hepatic and intestinal conditions [10, 15]. Furthermore, B. BORI demonstrated the maintenance of gut microbiota balance and stability by showing effectiveness in preventing and treating rotavirus infection in infants[16].

2) The author seems to have partially mentioned the number of experiments in the experiment method and results. It is desirable that the total number of experiments performed in this part be accurately reflected.

Response: Thank you for your comments. Osteoarthritis was induced in Wistar rat (n=6). The experiment was performed two times. We revised the Material and method section (marker by red color)

3) Why are some experiments used after killing B. longum BORI? An explanation is needed as to why dead bacteria were used.

Response: Heat-killed probiotics are reportedly safer than live probiotics for eliminating antibiotic-resistance genes, preventing the production of recombinant strains, and controlling the microbial load during probiotic supplementation (Ishibashi, N. & Yamazaki, S. Probiotics and safety. Am. J. Clin. Nutr. 73,465S–470S (2001); Vintini, E.O. & Medina, M.S. Host immunity in the protective response to nasal immunization with a pneumococcal antigen associated with live and heat-killed Lactobacillus casei. BMC Immunol. 12, 46 (2011)). Thus, we elected to use heat killed B. longum BORI in this study.

4) In methods, "Preparation of bacteria" it is not mentioned what sample was used for bacteria isolation

The authors would like to thank the comment of the reviewer. Following the reviewer’s comment, the authors revised the reference as follows: (marker by red color)

B. longum BORI was isolated from human feces of healthy infant who lived in Chuncheon, South Korea between 1995 and 1998, identified with 16S rRNA sequencing and cultured with MRS (De Man, Rogosa and Sharpe, Becton Dickson, Franklin Lakes, NJ, USA) broth medium at 37 ◦C. For the initial screening, lyophilized powder of each strain was used. Freeze-dried powder of B. longum BORI was provided by BIFIDO Co. Ltd (Hongchun, Korea)

5)In the same section, "Immunohistochemical staining of joint synovium showed that administration of B. longum BORI reduced the levels of IL-1b, LTB4r, PGE2, MMP9 and MMP13 (Figures 3 and 4)." . Histology in the method of Figures 3, 4 and 4 is not clear how to observe the reduction of the listed cytokines. It is necessary to add a description of how the cytokine reduction is observed.

Response: We agree and added the following text in method section. (marker by red color) 

The all IHC images were obtained each rat, and showing representative images. The percentage of positive cells showing in each image was measured by Image J.

6) The title refers only to experiments with rats. The experiment also includes the contents of the experiment in the patient's chondrocyte culture, so it would be good to mention these parts in the title.

Response: Thank you for your comment. We have revised the title to “

Bifidobacterium longum BORI reduces pain behavior and attenuates the progression of monosodium iodoacetate-induced osteoarthritis and 

Bifidobacterium longum BORI inhibits Pain behavior and chondrocyte death, and attenuates osteoarthritis progression.

7) In figure, B. longum BORI was not mentioned in Italics. It should be corrected throughout the Figure.

Response: Thank you for your comment. We revised the manuscript. 

8) The authors should better check the manuscripts for any typographical errors.

For example, LTB4 must be modified to LTB4r in figure 3.

Response: We apologize for this error, which has been corrected.

---

## [Decision Letter · Decision Letter 1]

17 May 2023

Bifidobacterium longum BORI inhibits Pain behavior and chondrocyte death, and attenuates osteoarthritis progression.

PONE-D-23-01788R1

Dear Dr. Cho,

We’re pleased to inform you that your manuscript has been judged scientifically suitable for publication and will be formally accepted for publication once it meets all outstanding technical requirements.

Kind regards,

Win Min Oo

Academic Editor

PLOS ONE

Additional Editor Comments (optional):

Reviewers' comments:

Reviewer's Responses to Questions

**Comments to the Author**

1. If the authors have adequately addressed your comments raised in a previous round of review and you feel that this manuscript is now acceptable for publication, you may indicate that here to bypass the “Comments to the Author” section, enter your conflict of interest statement in the “Confidential to Editor” section, and submit your "Accept" recommendation.

Reviewer #1: All comments have been addressed

2. Is the manuscript technically sound, and do the data support the conclusions?

Reviewer #1: Yes

3. Has the statistical analysis been performed appropriately and rigorously? 

Reviewer #1: Yes

4. Have the authors made all data underlying the findings in their manuscript fully available?

Reviewer #1: Yes

5. Is the manuscript presented in an intelligible fashion and written in standard English?

Reviewer #1: Yes

6. Review Comments to the Author

Reviewer #1: The authors submitted revised manuscript.

After review, previous recommandations were well responded and corrected.

I think this paper would be acceptable for publication.

With regards,

7. PLOS authors have the option to publish the peer review history of their article (what does this mean?). If published, this will include your full peer review and any attached files.

Reviewer #1: No

---

## [Editor Report · Acceptance letter]

13 Jun 2023

PONE-D-23-01788R1 

*Bifidobacterium longum* BORI inhibits Pain behavior and chondrocyte death, and attenuates osteoarthritis progression. 

Dear Dr. Cho:

I'm pleased to inform you that your manuscript has been deemed suitable for publication in PLOS ONE. Congratulations! Your manuscript is now with our production department. 

Kind regards, 

on behalf of

Dr. Win Min Oo 

Academic Editor

PLOS ONE